# MGMT Promoter Methylation: Prognostication beyond Treatment Response

**DOI:** 10.3390/jpm13060999

**Published:** 2023-06-14

**Authors:** Keyoumars Ashkan, Asfand Baig Mirza, Christos Soumpasis, Christoforos Syrris, Dimitrios Kalaitzoglou, Chaitanya Sharma, Zachariah Joseph James, Abbas Khizar Khoja, Razna Ahmed, Amisha Vastani, James Bartram, Kazumi Chia, Omar Al-Salihi, Angela Swampilai, Lucy Brazil, Ross Laxton, Zita Reisz, Istvan Bodi, Andrew King, Richard Gullan, Francesco Vergani, Ranjeev Bhangoo, Safa Al-Sarraj, Jose Pedro Lavrador

**Affiliations:** 1Kings College Hospital NHS Foundation Trust, Denmark Hill, London SE5 9RS, UK; asfand.mirza@nhs.net (A.B.M.); christos.soumpasis@nhs.net (C.S.); c.syrris@nhs.net (C.S.); dimitrios.kalaitzoglou@nhs.net (D.K.); a.vastani@nhs.net (A.V.); james.bartram2@nhs.net (J.B.); kazumi.chia2@nhs.net (K.C.); richardgullan@nhs.net (R.G.); francesco.vergani@nhs.net (F.V.); ranj.bhangoo@nhs.net (R.B.); safa.al-sarraj@nhs.net (S.A.-S.); josepedro.lavrador@nhs.net (J.P.L.); 2GKT School of Medicine, Kings College London, London SE1 1UL, UK; 1stvan1ah.1.sharma@kcl.ac.uk (C.S.); 1stvan1ah.james@kcl.ac.uk (Z.J.J.); khizar.khoja@kcl.ac.uk (A.K.K.); razna.ahmed@kcl.ac.uk (R.A.); 3Department of Neuro-Oncology, Cancer Centre, Guys Hospital, Great Maze Pond, London SE1 9RT, UK; omar.al-salihi@nhs.net (O.A.-S.); a.swampillai@nhs.net (A.S.); l.brazil@nhs.net (L.B.); 4Department of Neuropathology, Kings College London, London SE5 9RS, UK; ross.laxton@nhs.net (R.L.); zita.reisz@nhs.net (Z.R.); andrewking@nhs.net (A.K.)

**Keywords:** MGMT, methylation, GBM, glioblastoma, 5-ALA

## Abstract

MGMT promoter methylation is related to the increased sensitivity of tumour tissue to chemotherapy with temozolomide (TMZ) and thus to improved patient survival. However, it is unclear how the extent of MGMT promoter methylation affects outcomes. In our study, a single-centre retrospective study, we explore the impact of MGMT promoter methylation in patients with glioblastoma who were operated upon with 5-ALA. Demographic, clinical and histology data, and survival rates were assessed. A total of 69 patients formed the study group (mean age 53.75 ± 15.51 years old). Positive 5-ALA fluorescence was noted in 79.41%. A higher percentage of MGMT promoter methylation was related to lower preoperative tumour volume (*p* = 0.003), a lower likelihood of 5-ALA positive fluorescence (*p* = 0.041) and a larger extent of resection EoR (*p* = 0.041). A higher MGMT promoter methylation rate was also related to improved progression-free survival (PFS) and overall survival (OS) (*p* = 0.008 and *p* = 0.006, respectively), even when adjusted for the extent of resection (*p* = 0.034 and *p* = 0.042, respectively). A higher number of adjuvant chemotherapy cycles was also related to longer PFS and OS (*p* = 0.049 and *p* = 0.030, respectively). Therefore, this study suggests MGMT promoter methylation should be considered as a continuous variable. It is a prognostic factor that goes beyond sensitivity to chemotherapy treatment, as a higher percentage of methylation is related not only to increased EoR and increased PFS and OS, but also to lower tumour volume at presentation and a lower likelihood of 5-ALA fluorescence intraoperatively.

## 1. Introduction

Glioblastoma (GBM) is an aggressive primary malignancy of the central nervous system (CNS) [1], predominantly diagnosed in older populations with a median age of 64 [2,3]. It is the most common primary malignancy of the CNS, accounting for 14.5% of all CNS tumours and 48.6% of malignant CNS tumours [4,5]. The incidence rate is 4.6/100,000 people in England [6]. The current treatment protocol for GBM is maximal surgical resection, followed by concomitant radiotherapy (RT) and temozolomide (TMZ) [7,8,9]. Despite this treatment, the prognosis for patients with GBM is poor, with a median overall survival (OS) rate of just 15 months and a 3-year survival rate of only 16% [4,8,10].

TMZ is a cytotoxic agent that suppresses DNA replication in tumours. It acts by alkylating guanine at the O^6^ position in DNA, which leads to irrevocable mutations in DNA and ultimately to arrest in its replication. The enzyme O-6-methylguanine-DNA methyltransferase (MGMT) reverses alkylation and can reduce TMZ-induced cell death and gene mutation [11]. The expression of MGMT is controlled by the methylation at the 5′-cytosine-phosphate-guanine-3′ (CpG) islands along the MGMT gene promoter. Methylation of the MGMT gene promoter reduces the expression of the MGMT protein, allowing increased sensitivity of the tumour tissues to TMZ [12,13].

The techniques most commonly used to measure this methylation are methylation-specific PCR (MSP) and pyrosequencing (PSQ). MSR uses the reaction between bisulfite and cytosine to determine the methylation status of a DNA strand. If the cytosine residues are methylated, they will not react with the bisulfite. However, if they are unmethylated, they convert to uracil [14]. PSQ is a DNA sequencing technique that is based on the detection of bioluminescent signals released as nucleotides that are incorporated into a DNA strand. These bioluminescent signals are generated by the release of pyrophosphate during the serial addition of nucleotides by DNA polymerase. PSQ is a powerful technique that generates quantitative data regarding the extent of methylation of the MGMT promoter region and has proven to be more reliable than MSR [15,16,17,18].

MGMT methylation status has been proposed as a predictor of patient survival [19]. Numerous studies have been conducted that associate the methylation of the MGMT gene promoter region with better survival outcomes for those with GBM treated with TMZ [20,21,22,23,24,25]. Despite having highlighted MGMT methylation status as a positive prognostic factor, its clinical implementation is yet to be conclusively defined [26,27]. There is still no distinct cut-off defining the extent of MGMT methylation that characterizes methylated and unmethylated patient groups [28,29]. As a result, the threshold to differentiate a survival benefit within different studies ranges from 6 to 40% [29,30,31,32,33,34,35,36,37,38]. A significant confounding factor for this is the intratumoural heterogeneity in GBM, which may be particularly relevant in borderline and medium methylated tumours [36]. This raises the problem of sampling errors, which may lead to different treatment strategies in similar cohorts of patients. Moreover, it is not clear if MGMT promoter methylation status as a predictor of patients’ outcomes should be considered a dichotomic, ordinal or continuous variable. For example, Hegi et al. found that there may be no additional survival benefit after a particular degree of methylation of the MGMT promoter region [30].

The vast majority of the literature that assesses the impact of MGMT promoter methylation precedes the widespread use of 5-Aminolevulinic Acid (5-ALA) in glioblastoma surgery. 5-ALA positivity is potentially related to certain molecular characteristics, such as stronger fluorescence in IDH-wildtype tumours, or with non-neoplastic tissue in recurrent tumours [39,40]. However, no correlation thus far has been found between 5-ALA and MGMT promotor methylation status [39].

Our group published the positive impact of MGMT promoter methylation status as a dichotomic variable in a mixed 5-ALA and non-5-ALA-guided surgery cohort, where the adjusted model suggested a positive impact, even when 5-ALA-guided surgery was performed [41]. Recent data suggest that a residual tumour significantly and negatively impacts outcomes even in MGMT-methylated tumours [42]. The impact of MGMT methylation on surgical resection of tumours, however, remains unexplored.

In this paper, we explore the impact of MGMT promoter methylation in a population of GBM patients treated with 5-ALA-guided surgery.

## 2. Materials and Methods

This is a single-centre retrospective cohort study of consecutive patients admitted for surgery with glioblastomas. All patients in our practice with suspected high-grade gliomas, who undergo debulking surgery, receive 5-ALA. The inclusion criteria were adult patients (age > 18 years old), 5-ALA-guided surgery for maximal safe tumour resection and histologically confirmed GBM according to the WHO 2021 classification of CNS tumours [43]. The exclusion criteria were incomplete clinical and/or radiological information, tumours that were IDH mutant and reclassified as astrocytoma WHO Grade 4 and absent quantitative MGMT promotor methylation data.

The following clinical variables were collected: age, sex, performance status at presentation and follow-up, signs and symptoms at presentation, intraoperative 5-ALA fluorescence (positive vs. negative), volumetric extent of resection (EoR), postoperative oncological treatment, PFS and OS. PFS was calculated as the time between the date of surgery and the date of diagnostic MRI supporting disease progression. OS was calculated as the time between the date of diagnostic MRI before surgery (first surgery in case of reoperations) and the date of death. From an imaging perspective, we performed a volumetric assessment, using PACS SECTRA option, of the tumour at presentation and the residual volume after surgery. The EoR was calculated as the ratio between contrast-enhancing residual volume and tumour volume at presentation.

MGMT promoter methylation was assessed as a dichotomic variable with a cut-off of 9% [36], as an ordinal variable—non-methylated (0–9%), low methylated (10–17%), medium methylated (18–29%) and highly methylated (30–100%) [44]—and as a continuous variable. MGMT pyrosequencing was as follows: DNA was extracted from Fresh Frozen (FF) material using AllPrep DNA/RNA/miRNA Universal Kit (Qiagen/80224) or FFPE sample using AllPrep FFPE Kit (Qiagen/80234) and was processed on an automated Qiagen QIAcube platform (Model: Connect MDx/030048). Methylation status of the O6-methylguanine DNA methyltransferase (MGMT) gene promoter at four CpG sites (in exon 1; genomic sequence on chromosome 10 from 131,265,519 to 131,265,537: CGACGCCCGCAGGTCCTCG) was examined by pyrosequencing technology using the therascreen MGMT pyro kit (Qiagen) according to the manufacturer’s specifications. Pyrogram traces both from controls and examined samples were obtained after analysis of bisulfate-converted DNA and the average (%) methylation was calculated.

A literature review of 15 articles regarding the effect of MGMT promoter methylation on median OS for patients treated with TMZ after surgical resection was performed. Each study was segregated into two tables (Table A1 and Table A2) to display the median OS for patients with an unmethylated MGMT promoter versus patients with a methylated MGMT promoter. Seven studies used MSR to qualitatively measure the methylation of the MGMT promoter and the remaining studies used PSR to quantify the extent of methylation of the MGMT promoter region.

Stata 13.1 software package was used to perform statistical analysis. A *p* value < 0.05 was considered statistically significant. Different statistical approaches were performed according to the treatment, depending on the variable of interest. When MGMT promotor methylation was treated as a dichotomic variable, logistic regression was used; for ordinal organization of the variable, the analysis was performed with ordered logistic regression; for continuous variable, a linear regression was used. Kaplan–Meier curves were used for survival analysis as well as Cox hazard ratios to assess the impact of the treatment-related variables in the different MGMT promoter methylation subgroups.

## 3. Results

In total, 69 patients were included (51 males, 18 females, mean age 53.75 ± 15.51 years-old). In regard to performance status (PS), 52.17% (36/69) of the patients were PS0 and 42.03% (29/69) were PS1. The most common symptoms were headaches—44.93% (31/69), focal neurological deficit—19.12% (13/69), gait ataxia—18.84% (13/69) and seizures—17.39% (12/69). Lesion laterality was evenly distributed across the cohort. The most common tumour locations were frontal—31.88% (22/69) and temporal—30.43% (21/69).

All patients underwent 5-ALA-guided craniotomy for maximal safe resection after a neuro-oncology multi-disciplinary team assessment. Positive fluorescence with 5-ALA was noted in 79.41% (54/69) and 47.06% of the patients (32/69) were operated upon with intraoperative neuromonitoring. (Figure 1 and Figure 2) Gross total resection (GTR) was achieved in 55.07% (38/69) of the patients.

MGMT methylation status was related to the characteristics of the tumour at presentation. Despite not having a correlation with age (*p* = 0.684) or gender (*p* = 0.991, Logistic Regression), MGMT-methylated tumour patients had a lower PS at presentation (coef. −1.02 ± 0.49, 95%CI [−1.97–−0.06], *p* = 0.038). A higher MGMT methylation percentage was related to lower preoperative tumour volume (coef. −0.29 ± 0.09, 95%CI [−0.48–−0.11], *p* = 0.003). This was still the case when a binary classification of the MGMT methylation status was considered (MGMT unmethylated—33.98 ± 23.22cc3 versus MGMT methylated—22.62 ± 22.96cc3, *p* = 0.0482). A higher MGMT methylation percentage was related to a lower likelihood of intraoperative 5-ALA positive bright fluorescence (coef. −0.04 ± 0.02, 95%CI [−0.07–−0.001], *p* = 0.041). Again, this was still the case when the binary classification was considered (5-ALA positive MGMT unmethylated—90.9% versus 5-ALA positive MGMT methylated—68.6%, *p* = 0.023). Despite this impact on 5-ALA positive fluorescence, a higher degree of MGMT methylation was related to larger EoR (coef. 0.24 ± 0.11, 95%CI [0.01–0.47], *p* = 0.041). This effect was independent of the preoperative tumour volume (EoR was not related to preoperative tumour volume—*p* = 0.923), especially when adjusted for this variable (coef. 0.28 ± 0.12, 95%CI [0.04–0.53], *p* = 0.026). A binary classification of MGMT methylation did not demonstrate this impact (EoR in MGMT unmethylated patients—88.58 ± 20.30% versus EoR in MGMT-methylated patients—95.39 ± 11.67%, *p* = 0.1148) (Figure 3).

The vast majority of patients (75.86%) had concomitant radio-chemotherapy, followed by adjuvant chemotherapy with a variable number of cycles according to clinical tolerance; 12.07% had concomitant radio-chemotherapy and did not progress with chemotherapy due to side effects related to the concomitant phase of treatment.

PFS was significantly longer with a higher percentage of MGMT promoter methylation (HR 0.97 ± 0.01, 95%CI [0.96–0.99], *p* = 0.008, Cox hazard ratio) even when adjusted for EoR (HR 0.98 ± 0.01, 95%CI [0.97–0.99], *p* = 0.034, Cox hazard ratio). Similar results were verified with the OS (HR 0.97 ± 0.01, 95%CI [0.95–0.99], *p* = 0.006, Cox hazard ratio), even when we adjusted the results to the EoR (HR 0.80 ± 0.009, 95%CI [0.65–0.99], *p* = 0.042, Cox hazard ratio) (Table 1 and Figure 4).

A subgroup analysis focusing on the impact of the number of TMZ cycles on the PFS and OS according to the percentage of MGMT promoter methylation was performed. The subgroup of patients with only radio-chemotherapy was too small for a meaningful statistical assessment. When patients received adjuvant chemotherapy with TMZ following radio-chemotherapy, a higher amount of MGMT promoter methylation (HR 0.97 ± 0.01, 95%CI [0.95–0.99], *p* = 0.036) and a higher number of TMZ cycles (HR 0.78 ± 0.09, 95%CI [0.63–0.98], *p* = 0.030) were related to longer OS. A similar effect was identified with PFS (MGMT promoter methylation—HR 0.97 ± 0.01, 95%CI [0.96–0.99], *p* = 0.038; TMZ cycles—HR 0.84 ± 0.01, 95%CI [0.71–0.99], *p* = 0.049).

## 4. Discussion

This study demonstrates that MGMT promoter methylation is more than a dichotomic variable in patients with GBM. This is related not only to the extent of response to chemotherapy, but also to different tumour characteristics. A higher percentage of methylation is related to lower tumour volume at presentation, a lower likelihood of intraoperative 5-ALA fluorescence and an increased EoR. From a prognostication perspective, a higher degree of MGMT promoter methylation is related to increased PFS and OS, even when adjusted for the EoR beyond the minimal cut-off for methylation.

The survival benefit associated with a silenced MGMT gene was first described by Hegi et al. in 2005 [21]. Since then, MGMT promoter methylation has persistently been demonstrated as a positive prognostic factor, by virtue of conferring a better response to treatment, on a binary scale. Hegi et al. showed the prognostic significance of silencing MGMT regardless of the type of treatment [21]. Patients were treated with either radiotherapy only or with chemoradiotherapy and adjuvant chemotherapy. A significant difference in survival was identified between methylated and unmethylated patients in both treatment groups. A difference of 3.5 months and 9 months was found in median OS in the radiotherapy and chemoradiotherapy treatment groups, respectively. As shown in Table A1 and Table A2, a similar trend can be seen in other studies with a range of differences in median OS between 1.0 and 1.7 months for the radiotherapy only treatment groups and 2.7–13.0 months for the groups treated with chemo-radiotherapy and adjuvant chemotherapy [19,20,21,22,23,24,33].

Although at first glance these findings suggest MGMT methylation as a marker of better response to treatment, we propose a more fundamental difference and hypothesize a potential intrinsic difference in the tumour biology related to the MGMT promoter methylation status that confers a better prognosis per se, irrespective of the treatment offered to the patient. This view is further supported by our study, where the prognostic value of MGMT methylation appears to go beyond a differential response to treatment between the methylated versus unmethylated tumours, to reflect a basic difference in the biology of these two variants of the tumour. Thus, a higher degree of MGMT promoter methylation was related to better performance status at presentation, lower initial tumour volume and a higher extent of resection. Even though we could assume that the higher EoR was related to the lower preoperative tumour volume, these variables were proven not to be related. In addition, the magnitude of effect of the MGMT methylation status in the EoR was increased when adjusting for preoperative tumour volume.

Our study demonstrates that the presence of 5-ALA fluorescence is also potentially related to MGMT methylation status. Specchia et al. investigated the correlation of the fluorescence expression with different molecular characteristics of GBM. Their study noted a statistically non-significant trend of strong fluorescence in MGMT promoter un-methylated GBMs [39]. Jaber et al., on the other hand, did not find any correlation between MGMT promoter methylation status and the degree of 5-ALA fluorescence [45]. The lower extent of fluorescence in MGMT-methylated patients in our cohort may further support their lower biological aggressivity given that stronger fluorescence is known to be associated with increased malignancy in GBM [41].

Our results in regard to MGMT promoter methylation and PFS/OS are aligned with the previous published literature (Table A1 and Table A2). However, we go beyond this dichotomic association and show a progressive positive impact of the degree of methylation in the PFS and OS, both when the different categories are considered (non, low, medium and highly methylated) and when it is considered as a continuous variable. Additionally, the positive relationship remains significant when the results are adjusted for the confounding factor of the EoR.

In the subgroup of patients who progressed to chemo-radiotherapy, a higher percentage of MGMT promoter methylation and a higher number of TMZ cycles were related to increased OS and PFS. Although speculative, these results may suggest the use of extended cycles of temozolamide, beyond the usual six cycles, in this group of patients [46,47]. However, more cycles maybe associated with a higher risk of side effects [47]. Therefore, further clinical studies are required to evaluate this.

### Limitations and Strengths

This is a retrospective case series affected by missing data points after interrogation of electronic patient records. Moreover, the treatment decision taken for this group of patients took into consideration a dichotomic definition of MGMT promoter methylation and not the concept we assessed in this project—the impact of continuous MGMT promoter methylation. Additionally, out of the 87.93% of the patients for whom there was an initial intention to treat with the Stupp protocol, only 75.86% progressed beyond chemo-radiotherapy towards adjuvant chemotherapy. Therefore, the PFS and OS outcomes have to take into consideration the bias of the initial treatment decision based on a different definition from the one we are assessing and the selection bias of the patients who progressed through the chemoradiotherapy towards adjuvant chemotherapy only. Although further chromosomic analysis assessing the impact of MGMT promoter methylation was not performed in this study, such as the evaluation of microsatellite instability, we believe this should be the scope of future research to attempt a parallelism with what happens in other systemic conditions, such as colorectal cancer. Nevertheless, this paper provides a single-centre multidisciplinary team-based retrospective cohort assessing a large number of patients where continuous MGMT promoter methylation data were available in the 5-ALA era. This work adds further information to the ongoing discussion about treatment stratification and outcome measures based on MGMT. These preliminary results, showing the value of MGMT beyond the current perspective, allow for further larger studies to be performed to validate our hypothesis.

## 5. Conclusions

MGMT promoter methylation is more than a dichotomic prognostic factor related to response to chemotherapy with temozolamide. This study suggests it is a continuous measure that can be used to further assist in prognostication. In addition, its influence extends way beyond the response to chemotherapy as a higher percentage of methylation is related to such basic biological parameters such as lower tumour volume at presentation and a lower likelihood of 5-ALA fluorescence intraoperatively. We suggest MGMT-methylated tumours represent a less aggressive variant of GBM with a better prognosis.

## Figures and Tables

**Figure 1 jpm-13-00999-f001:**
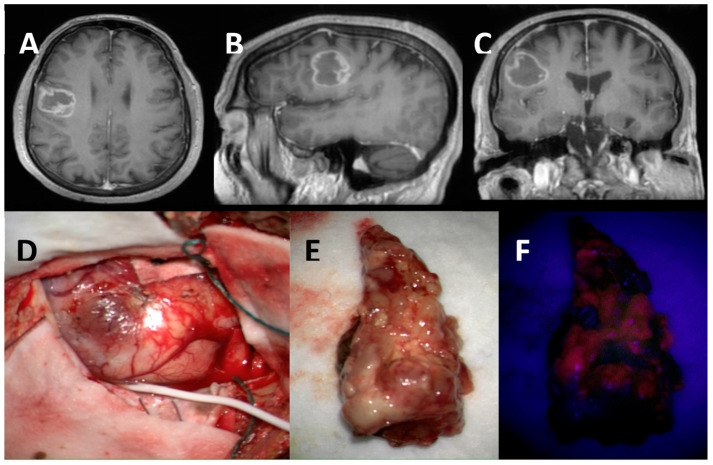
Highly methylated MGMT promoter methylation (37.93%): Axial (**A**), Sagittal (**B**) and Coronal (**C**) views of T1-Gadolinium weighted image showing a right frontal contrast-enhancing lesion in the ventral premotor cortex. Intraoperative image of the tumour invading the cortical surface (subdural strip in place for intraoperative neuromonitoring) (**D**). Tumour specimen under white light (**E**) and BLUE 400 Filter showing non-homogenous moderate 5-ALA fluorescence (**F**).

**Figure 2 jpm-13-00999-f002:**
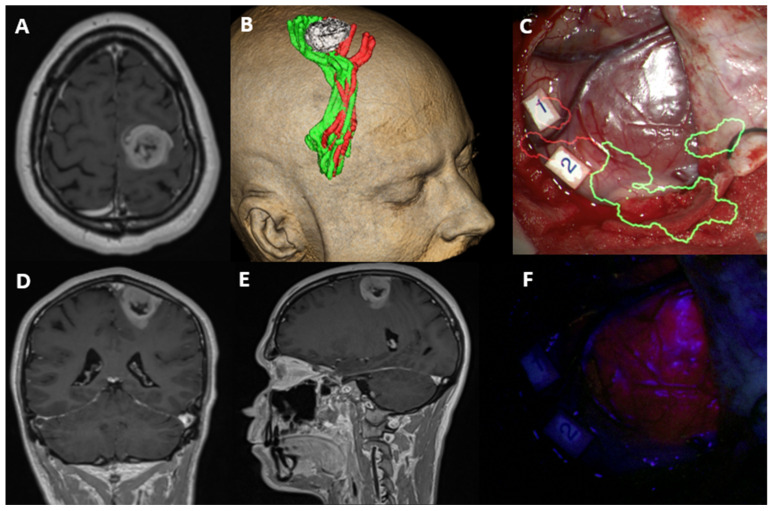
Low methylated MGMT promoter methylation (4.5%): Axial (**A**), Coronal (**D**) and Sagittal (**E**) views of T1-Gadolinium weighted image showing a right frontal contrast-enhancing lesion within the primary motor and sensory cortices. 3D reconstruction of the tumour (white) and lower limb (green) and upper limb (red) corticospinal tract fibres embracing the lesion (**B**). Intraoperative image of the tumour invading the cortical surface with augmented reality for the corticospinal tract fibres (**C**). Tumour visualized under white light (**C**) and BLUE 400 filter showing homogenous and bright fluorescence (**F**). Number 1 and 2 are cortical areas with positive motor activation for upper limb with high frequency stimulation paradigm.

**Figure 3 jpm-13-00999-f003:**
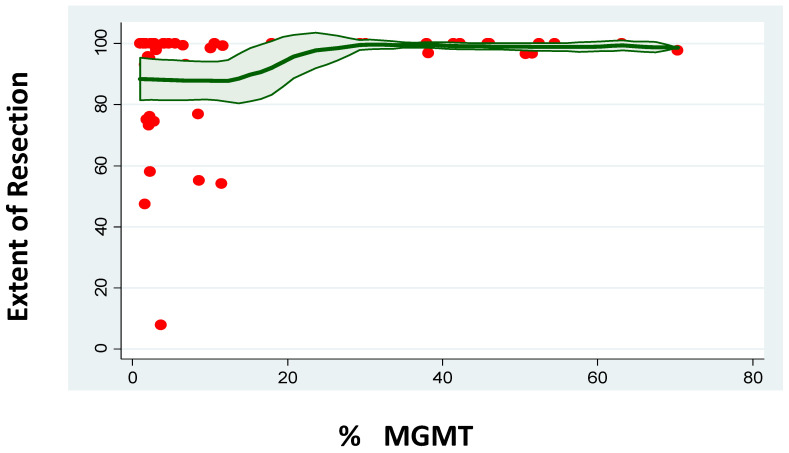
Correlation between percentage of MGMT promoter methylation and the extent of resection.

**Figure 4 jpm-13-00999-f004:**
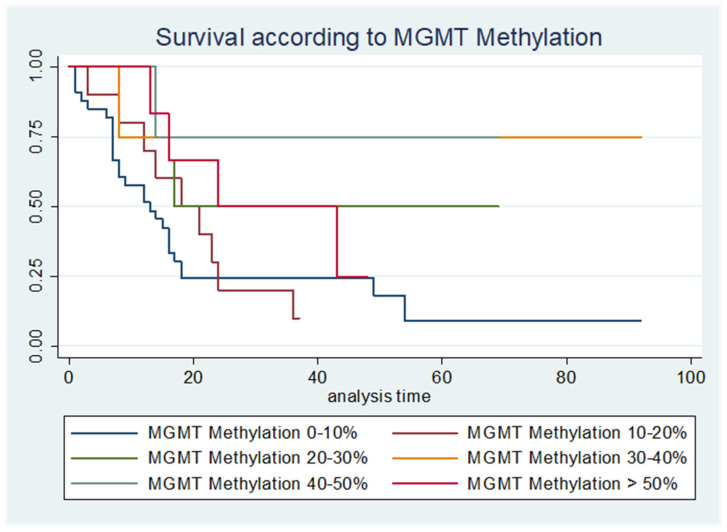
Kaplan–Meier curves illustrating the different survival curves according to different categories of MGMT promotor methylation, *p* = 0.008.

**Table 1 jpm-13-00999-t001:** Demographics of our cohort in terms of MGMT promotor methylation, overall survival(OS) and PFS (Performance status).

	PFS	OS	n
MGMT Promoter Methylation 0–10% (months)	9	13	34
MGMT Promoter Methylation 10–20% (months)	16	18	10
MGMT Promoter Methylation 20–30% (months)	19	17	6
MGMT Promoter Methylation 30–40% (months)	31	-	4
MGMT Promoter Methylation 40–50% (months)	37	-	4
MGMT Promoter Methylation <50% (months)	21	24	6

(mean OS has not been achieved for MGMT Promotor Methylation Groups 30–40% and 40–50%).

## Data Availability

Data availability is available upon request.

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
