# Peer review of "MGMT Promoter Methylation: Prognostication beyond Treatment Response"

_jpm, 2023, doi:10.3390/jpm13060999_

Round 1

Reviewer 1 Report

In this paper the author present a retrospective case series on the prognostic influence of MGMT methylation in GBM patients treated with 5-ALA guided resection followed by Stupp protocol.

They found a significant correlation between MGMT methylation status, intended as a continuous variable, and the EoR, PFS and OS. They conclude that MGMT methylation status should be considered a continuos parameter that intrinsically profiles a lesser aggressive variant of GBM and could be used for prognostication.

The paper is well written and easy to understand, english is fine. Auto-citation rate is low (1/51). Conclusions are sound and coherent with premises.

Authors are advised to:

-carefully check for typo-errors (see line 6,13,253)

-add a quality-checklist from the EQUATOR network (see STROBE statement)

Author Response

Dear reviewer, thank you for your feedback:

-carefully check for typo-errors (see line 6,13,253)

Response 1: We have gone through the manuscript and corrected the typos.

-add a quality-checklist from the EQUATOR network (see STROBE statement)

Response 2: We have added the STROBE checklist.

Reviewer 2 Report

Very good study with potential practical clinical implication. I have just a few suggestions to clarify / improve the paper for the readers:

1. Please include images/photos for the intraoperative 5-ALA fluorescence that could potentially differentiate groups of patients and that corrrelates to the degree of MGMT methylation.

2. The relation of MGMT and MSI is known at least in colorectal cancer.  Did you study the MSI status for these patient? What was the correlation with MGMT methylation status in GBM?

Author Response

1. Please include images/photos for the intraoperative 5-ALA fluorescence that could potentially differentiate groups of patients and that corrrelates to the degree of MGMT methylation.

Response: Thank you for the suggestion of adding images, we have added 2 images capturing intraoperative 5-ALA fluorescence in both high and low Methylated MGMT Promoter Methylation tumours. We hope this suffices.

2. The relation of MGMT and MSI is known at least in colorectal cancer.  Did you study the MSI status for these patient? What was the correlation with MGMT methylation status in GBM?

Response: 

Thank you very much for your comment. As you mentioned, microsatellite instability is well-known in other tumors such as colorectal cancer.  We believe this is a very interesting line of research and should be the scope of future studies. Unfortunately, we do not have the data to perform the correlation of MSI and MGMT in this study.  We have addressed this in the limitation section.   "Further chromosomic analysis assessing the impact of MGMT promotor methylation were not performed in this study, such as the evaluation of microsatellite instability. We believe this should be the scope of future research to attempt a parallelism to what happens in other systemic conditions, such as colerectal cancer."